**Data Availability Statement:** The data underlying the results presented in the study are freely available from the DataverseNL repository at:

# Assessing the reliability and validity of the Slovenian version of the Appraisal of Diabetes Scale (ADS-S) in type 2 diabetes patients

Matic Mihevc[1,2]*, Špela Miroševič[2], Majda Mori Lukančič[1], Tina Virtič Potočnik[1,3], Črt Zavrnik[1,2], Marija Petek Šter[2], Zalika Klemenc-Ketiš[1,3], Antonija Poplas Susič[1,2]

1 Primary Healthcare Research and Development Institute, Community Health Centre Ljubljana, Ljubljana, Slovenia, 2 Department of Family Medicine, Faculty of Medicine, University of Ljubljana, Ljubljana, Slovenia, 3 Department of Family Medicine, Faculty of Medicine, University of Maribor, Maribor, Slovenia

* mihevc.matic@zd-tr.si

## Abstract

Managing type 2 diabetes (T2D) effectively is a considerable challenge. The Appraisal of Diabetes Scale (ADS) has proven valuable in understanding how individuals perceive and cope with their condition. This study aimed to evaluate the psychometric properties of the Slovenian version of ADS (ADS-S). We recruited a sample of 400 adult individuals with T2D from three primary healthcare centers in Slovenia, ensuring an average of 57 cases per individual item. The psychometric evaluation included internal consistency, test-retest reliability, construct validity, and discriminant validity. Confirmatory factor analysis (CFA) was additionally performed to evaluate the fit of one- and two-factor models. After excluding incomplete questionnaires, 389 individuals participated, averaging 72.0±7.5 years, with 196 men and 193 women. ADS-S exhibited acceptable internal consistency (Cronbach's α = 0.70) and strong test-retest reliability (interclass correlation = 0.88, p <0.001). Criterion validity was established through significant correlations between ADS-S score and EQ-5D utility score (r = -0.34, p <0.001), EQ-VAS score (r = -0.38, p <0.001), and HbA1c >7.5% (r = 0.22, p = 0.019). Discriminant validity assessment found no significant correlation between ADS-S score and age, but a significant correlation with female gender (r = 0.17, p = 0.001). CFA results supported a two-factor structure (psychological impact of diabetes and sense of self-control) over a one-factor structure, as indicated by model fit indicators. ADS-S stands as a valid and reliable tool for assessing psychological impact and self-control in Slovenian T2D patients. Future research should explore adding items for capturing secondary appraisal of diabetes and studying the influence of female gender on ADS scores.

## Introduction

Type 2 diabetes (T2D) is widely recognized as a significant public health problem due to its profound impact on individual well-being and healthcare costs [1]. It causes functional limitations and impairs overall quality of life, often leading to severe morbidity and premature death

https://dataverse.nl/dataset.xhtml?persistentId=doi:10.34894/SPGBBR.

**Funding:** The research is financially supported by the SCUBY project, an international research initiative co-financed by the European Union through the H2020 - Health Program (H2020-SC1) and identified by contract number 825432 - SCUBY. SM acknowledges support from the Slovenian Research Agency ARIS (program P3-0339). The funders had no role in study design, data collection and analysis, decision to publish, or preparation of the manuscript.

**Competing interests:** The authors have declared that no competing interests exist.

[1–3]. Despite medical advances, many patients still struggle to effectively manage crucial target values such as blood glucose (BG) levels and blood pressure [4–6].

A key factor contributing to the suboptimal management of T2D is reliance on rigid treatment guidelines that advocate incremental addition of medications to control BG [7]. However, more recent guidelines emphasize the importance of adopting a personalized approach to improve patient satisfaction, quality of life, medication adherence, and overall health outcomes [7, 8]. This involves considering patient-specific factors such as treatment preferences, age, duration of T2D, fear of hypoglycemia, psychological and social circumstances [7, 8].

In clinical practice, a comprehensive questionnaire is essential to effectively screen patients for diabetes-related fears and identify those at risk of non-compliance with their treatment regimens. The Appraisal of Diabetes Scale (ADS) is a succinct, seven-item questionnaire derived from the stress and coping literature, designed to gain profound insights into individuals' perceptions and coping mechanisms related to their diabetes [2, 9]. The ADS captures crucial factors such as distress, control over diabetes, uncertainty, likelihood of deterioration, coping strategies, and interference with life goals [9].

The ADS questionnaire, initially introduced by Carey et al. in 1991, underwent its first validation exclusively among a male population diagnosed with T2D [9]. Subsequently, it was translated and validated in Japanese [10] and Korean [11], extending its reach to encompass both men and women with T2D. However, these studies have not arrived at a consensus regarding the questionnaire's underlying factor structure.

In the original study, a single factor comprising seven items was identified [9]. However, in a subsequent Japanese study, the questionnaire revealed a more complex structure with three distinct factors [10]. These factors encompassed aspects related to the psychological impact of diabetes, the sense of self-control, and the effects of symptom management [10]. The Korean study yielded findings consistent with the Japanese study but excluded the third factor, attributing it to a floor effect [11]. Given these conflicted results, it is crucial to acknowledge that perceptions of illness or health are strongly influenced by social norms, values, and environmental interactions [12]. Therefore, there may be a degree of variability in the applicability of ADS questionnaire depending on the contextual factors of different settings.

Slovenia, a high-income country in central Europe, has made significant efforts to implement an integrated care package for individuals with T2D in primary care settings [13–15]. Despite these efforts, challenges persist in providing adequate support for T2D self-management and empowering patients [5, 13, 14]. Individuals living with T2D in Slovenia frequently encounter personal difficulties when attending referral clinics and face barriers related to their personal motivation to adopt a healthy lifestyle or fully embrace their diagnosis [16]. Furthermore, there has been a worrying stagnation or deterioration in the older population's knowledge of how to manage diabetes over the past decade [17]. Given these challenges, it is critical to have an acceptable and validated tool that can effectively identify individuals who need additional support and intervention.

The aim of this study was to determine the psychometric properties of a Slovenian version of the ADS using a sample of Slovenian people with T2D.

## Materials and methods

### Study design

We conducted a multi-center cross-sectional study in the Slovenian adult population of people with T2D. The study was approved by the Medical Ethics Committee of the Republic of Slovenia (reference number 0120-219/2019/4) and compiled with Declaration of Helsinki. Written informed consent was obtained from participants upon inclusion in the study.

## Study setting

The study took place in three primary healthcare centers (PHCs) in Slovenia, reflecting different development contexts. PHC Ljubljana, with approximately 300,000 residents, showcased an urban profile, with a gross domestic product (GDP) of 108% of the EU average in 2021. Conversely, PHC Trebnje and PHC Slovenj Gradec, serving about 50,000 residents in Eastern Slovenia, reflected a rural context, with a GDP of 74% of the EU average in 2021 [18].

## Study population

The study included 400 people who met the inclusion criteria and were selected from three PHCs in Slovenia between June 1, 2022, and June 1, 2023. After exclusion of incomplete questionnaires, the final sample size was 389 patients.

Inclusion criteria were: (a) ≥18 years of age and (b) confirmed T2D diagnosis with fasting BG value ≥7.0 mmol/l or venous plasma glucose ≥11.1 mmol/l 2 hours after 75 g glucose tolerance test or any random opportunity. Patients with cognitive decline or other diabetes types, such as type 1 or gestational diabetes, were excluded.

## Sampling strategy and sample size determination

The sampling strategy was convenient until the planned sample size was reached. Patients were recruited by their primary care physicians during their routine visits to the PHCs.

The sample size determination followed previous validation studies on ADS involving up to 350 individuals with diabetes [9–11]. Adhering to prevalent suggestions for sample size in confirmatory factor analyses [19, 20], we opted for a larger sample size of 400 individuals. This larger sample size ensured a more comprehensive coverage of 57 cases per each item within the analysis, enhancing the statistical power and results reliability.

## Measures

**Sociodemographic profile.** Sociodemographic information was obtained from participants using a questionnaire that included information on age, gender, education, and employment status. When accessible, the HbA1c status from the previous month was retrieved from medical records.

**Appraisal of Diabetes Scale.** The ADS is a self-reported questionnaire used to determine the patient's appraisal of their diabetes. It includes seven items, each rated on a 5-point Likert scale (S1 Appendix). Two items (2 and 6) are scored in reverse order. The total score is the sum of the scores for all items and can range from 7 to 35, with a higher score indicating a more negative appraisal of diabetes [9].

The translation and adaptation of the ADS scale into Slovenian followed the guidelines of the World Health Organization and the International Society for Pharmacoeconomics and Outcomes Research [21]. The following steps were followed: (a) obtaining permission from the original author of the scale [9] to translate ADS into Slovenian, (b) forward translation of the scale into Slovenian by two independent experts, (c) reconciliation of the forward translations into a single forward translation marked as the ADS-S, (d) back translation of the ADS-S into English by two bilingual experts, (e) harmonization of all new translations with each other and with the initial version, (f) proofreading and finalization of the ADS-S (S2 Appendix).

**EQ-5D-5L and EQ-VAS questionnaires.** To assess health-related quality of life, we used the EQ-5D-5L questionnaire, which has been validated for Slovenian use [22] and consists of the EQ-5D descriptive system and the EQ visual analogue scale (EQ VAS).

The EQ-5D descriptive system covers five key dimensions: mobility, self-care, usual activities, pain/discomfort, and anxiety/depression. Respondents rate difficulty or discomfort on five-response scale ranging from "no problems" to "extreme problems." These produce weighted EQ-5D utility scores ranging from -1.09 (lowest health status) to 1.000 (highest health status). A specific EQ-5D-5L value set for Slovenia was used in transformation in our data [22]. Cronbach's alpha in our study was 0.85.

The EQ VAS uses a visual analogue scale from 0 to 100 for individuals to self-evaluate their health status. A mark on the scale reflects their subjective well-being, with 0 representing the poorest health and 100 representing optimal health [22].

## Data analysis strategy

We used SPSS Software version 25 for Windows (SPSS, Chicago, IL, USA) to conduct the statistical analysis. However, for model fit assessment in confirmatory factor analysis (CFA), we utilized STATA Software version 14 (STATA Corp, College Station, TX).

**Assessment of factor structure.** To explore the ADS-S factor structure, we examined the original one-factor model [9] and the Korean two-factor model [11], excluding the Japanese three-factor model due to concerns about treating a single item as an individual factor [10]. We chose CFA over exploratory factor analysis, as prior studies had already identified the factor structure of the ADS, and we aimed to validate a previously proposed factor structure rather than explore entirely new patterns within the data.

Firstly, we compared factor loadings and reliabilities for identical items in one- and two-factor structures. Subsequently, we assessed model fit using several fit indices, including Tucker-Lewis Index (TLI) $\geq 0.95$, Comparative Fit Index (CFI) $\geq 0.95$, Root Mean Square Error of Approximation (RMSEA) $\leq 0.06$, and Standardized Root Mean Squared Residual (SRMR) $\leq 0.08$ [23].

**Assessment of internal consistency and test-retest reliability.** To assess internal consistency, we used the Cronbach's $\alpha$ coefficient, aiming for a value $>0.7$, which is considered minimally acceptable, and ideally $>0.9$ [24].

For test-retest reliability, individuals completed the questionnaire again after a 14-day interval. Of the initial participants, 49 responded to the second round of questioning. Test-retest reliability was assessed by computing the interclass correlation coefficient (ICC) between the initial and subsequent measurements, using a mixed two-way approach with absolute agreement and the paired t-test for dependent samples. The ICC results followed established standards: $<0.5$ indicated low reliability, 0.5 to 0.75 showed moderate reliability, 0.75 to 0.90 suggested good reliability, and $>0.90$ denoted excellent reliability [25].

**Assessment of construct and discriminant validity.** We assessed criterion-related validity by computing the Pearson correlation coefficient between the ADS-S score and either the EQ-5D-5L index score, EQ-VAS score, or poorly controlled HbA1c status (i.e., HbA1c $>7.5\%$). We anticipated a negative correlation between the ADS-S score and both the EQ-5D-5L utility score and EQ-VAS score, and a positive correlation between the ADS-S score and poorly controlled HbA1c status.

For discriminant validity, we examined the Pearson correlation coefficient between the ADS-S score, gender, and age. Our hypothesis expected no significant correlation between age or gender and the ADS-S score. These relationships were assessed according to the Pearson coefficient's strength, categorized as weak (0.1–0.3), moderate (0.4–0.6), or strong (0.6–0.9), following Dancey & Reidy's classification [26]. Furthermore, differences in responses between females and males to the items were examined using an unpaired samples t-test.

## Results

### Sample characteristics

The study included 389 people with T2D, the majority of whom were >65 years old, had secondary school education, were retired, and had an HbA1c <7.5% (Table 1). The mean EQ-5D utility score corresponded to the average Slovenian population norm, which is 0.81 [27].

Participants faced diverse challenges across all EQ-5D dimensions. Specifically, 205 individuals (52.7%) encountered mobility issues, 69 (17.7%) experienced difficulties in self-care, 124 (31.9%) encountered hindrances in usual activities, 262 (67.4%) reported pain or discomfort, and 134 (34.4%) dealt with various levels of depression or anxiety.

### Individual item characteristics

The mean scores of the items ranged from 1.86 to 2.83. Items 1, 3, 5, and 7 represented the floor effect, with over 15% of respondents scoring the lowest for the respective item (Table 2).

### Internal consistency

The internal consistency of the ADS-S was acceptable but not optimal, yielding a Cronbach's alpha of 0.70. Table 2 displays individual item analysis, revealing that item 2 displayed the highest alpha if deleted (0.72), while item 7 showed the lowest alpha if deleted (0.60).

### Confirmatory factor analysis

In confirmatory factor analysis, a two-factor model performed more favorably with higher factor loadings, better internal consistency, with lower SRMR, RMSEA, and higher TLI and CFI values compared to the one-factor model (Tables 3 and 4).

**Table 1. Overview of socio-demographic and clinical characteristics.**

| | |
|---|---|
| **Sex** | |
| Male, n (%) | 196 (50.4) |
| Female, n (%) | 193 (49.6) |
| **Age**, years, mean (SD), n = 389 | 72.0 (7.5) |
| **Highest education achieved** | |
| Primary school, n (%) | 74 (19.0) |
| Secondary school, n (%) | 223 (57.3) |
| Vocational school, n (%) | 56 (14.4) |
| Bachelor's degree, n (%) | 30 (7.7) |
| Master's degree, n (%) | 6 (1.5) |
| **Employment status** | |
| Employed, n (%) | 22 (5.7) |
| Unemployed, n (%) | 1 (0.3) |
| Retired, n (%) | 366 (94.1) |
| **Clinical profile** | |
| HbA1c value, mean (SD), n = 120 | 7.1 (1.0) |
| EQ-5D index, mean (SD), n = 387 | 0.8 (0.2) |
| EQ-VAS, mean (SD), n = 387 | 71.4 (19.3) |
| ADS, mean (SD), n = 389 | 16.2 (3.6) |

n, number; SD, standard deviation; HbA1c, glycated hemoglobin; EQ-5D, EuroQol-5-dimension; EQ-VAS, EuroQol-Visual Analogue Scale; ADS, Appraisal of Diabetes Scale.

**Table 2. Appraisal of Diabetes Scale (ADS-S): Descriptive statistics, mean, standard deviation, floor, and ceiling effect of each single item.**

| Items | Responses, n (%) | | | | | Mean (SD) | Floor effect (%) | Ceiling effect (%) | Alpha if item deleted |
|---|---|---|---|---|---|---|---|---|---|
| | Not | Slight | Moderate | Very | Extreme | | | | |
| Item 1: Upsetting | 120 (30.9) | 154 (39.7) | 93 (24.0) | 18 (4.6) | 3 (0.8) | 2.05 (0.90) | 30.9 | 0.8 | 0.64 |
| Item 2: Sense of control | 47 (12.1) | 102 (26.3) | 181 (46.6) | 45 (11.6) | 13 (3.4) | 2.68 (0.95) | 12.1 | 3.4 | 0.72 |
| Item 3: Uncertainty | 169 (43.6) | 124 (32.0) | 77 (19.8) | 17 (4.4) | 1 (0.3) | 1.86 (0.90) | 43.6 | 0.3 | 0.63 |
| Item 4: Deterioration likelihood | 28 (7.2) | 141 (36.3) | 175 (45.1) | 43 (11.1) | 1 (0.3) | 2.61 (0.79) | 7.2 | 0.3 | 0.66 |
| Item 5: Self-control | 66 (17.0) | 162 (41.8) | 144 (37.1) | 12 (3.1) | 4 (1.0) | 2.29 (0.82) | 17.0 | 1.0 | 0.71 |
| Item 6: Coping | 20 (5.2) | 90 (23.2) | 220 (56.7) | 53 (13.7) | 5 (1.3) | 2.83 (0.77) | 5.1 | 1.3 | 0.69 |
| Item 7: Goal interference | 169 (43.6) | 114 (29.4) | 85 (21.9) | 19 (4.9) | 1 (0.3) | 1.89 (0.93) | 43.5 | 0.3 | 0.60 |

n, number; SD, standard deviation.

## Test-retest reliability

The ADS-S questionnaire displayed good test-retest reliability (ICC = 0.88, p<0.001) with no significant difference between the first and second measurement means (t = 1.25, p = 0.22). Pearson correlation indicated a high correlation between time 1 and time 2 (r = 0.79, p<0.001).

## Criterion validity

The ADS-S questionnaire demonstrated significant, moderate negative correlations with both EQ-5D utility score (r = -0.336, p<0.001) and EQ-VAS score (r = -0.380, p<0.001). All EQ-5D dimensions exhibited weak or moderate positive correlations, with the anxiety/depression dimension showing the highest correlation (r = 0.326, p<0.001). Additionally, a significant weak positive correlation was observed with poorly controlled HbA1c status (r = 0.215, p = 0.019).

## Discriminant validity

In terms of discriminant validity, the ADS-S questionnaire displayed a non-significant weak positive correlation with age (r = 0.07, p = 0.169) and a significant weak positive correlation with female gender (r = 0.170, p = 0.001). Table 5 revealed substantial differences in the mean ADS-S scores between men and women.

**Table 3. Factor loadings and reliabilities for the same items under two different models.**

| Item | Model 1 (One-factor) | | | Model 2 (Two-factor) | | |
|---|---|---|---|---|---|---|
| | Factor structure | Factor loading | Alpha | Factor structure | Factor loading | Alpha |
| Item 1: Upsetting | Factor 1 | 0.767 | 0.70 | Factor 1: psychological impact of diabetes | 0.808 | 0.80 |
| Item 3: Uncertainty | | 0.275 | | | 0.838 | |
| Item 4: Deterioration likelihood | | 0.798 | | | 0.629 | |
| Item 7: Goal interference | | 0.639 | | | 0.865 | |
| Item 2: Sense of control | | 0.372 | | Factor 2: sense of self-control | 0.860 | 0.64 |
| Item 6: Coping | | 0.335 | | | 0.860 | |
| Item 5: Self-control | | 0.849 | | Excluded | / | / |

**Table 4. Model fit comparison between a one-factor and two-factor models.**

| Model | $\chi^2$ (df) | SRMR | RMSEA | TLI | CFI |
|---|---|---|---|---|---|
| Model 1 (One-factor) | 128.989 (14), p <0.001 | 0.096 | 0.145 | 0.709 | 0.806 |
| Model 2 (Two-factor) | 32.638 (13), p = 0.002 | 0.065 | 0.062 | 0.946 | 0.967 |

$\chi^2$, Chi-Square; df; degrees of freedom; SRMR, Standardized Root Mean Square Residual; RMSEA, Root Mean Square Error of Approximation; TLI, Tucker-Lewis Index; CFI, Comparative Fit Index.

## Discussion

### Principal findings and comparison with the existing literature

This study provides valuable insights into the psychometric properties of the Slovenian version of the ADS questionnaire. Our findings indicate that the questionnaire exhibits acceptable reliability and validity, with its individual items demonstrating varying levels of discrimination among individuals in the general population with T2D.

The ADS-S questionnaire exhibited acceptable internal consistency as reflected by a Cronbach's alpha of greater than 0.7. Nevertheless, it fell short of the ideal value of greater than 0.9, which is consistent with the results of previous studies in which the Cronbach's alpha ranged from 0.7 to 0.8 [9–11]. This discrepancy could be due to several factors, such as a limited number of items, skewed responses (items 1, 3, 5, and 7 showing a floor effect), the presence of items with reversed scoring, and possible redundancy of items, especially item 5, which could not be assigned to any specific factor [24].

Rooted in Lazarus and Folkman's stress coping theory, the appraisal of diabetes is under debate regarding its unidimensional or multidimensional nature, as indicated by previous studies [9–11]. The appraisal of diabetes assumes two primary dimensions. The primary appraisal gauges the significance of an event, classifying it as irrelevant, positive, or stressful, thus impacting well-being. On the other hand, secondary appraisal investigates various coping strategies involving physical, social, psychological, and material resources. This secondary appraisal is pivotal, determining the overall outcome by assessing the possibility of action regarding the event and identifying risks [28, 29].

In the CFA, we explored three scenarios of how appraisal of diabetes could be understood: as a one-dimensional, two-dimensional, or three-dimensional construct. We excluded the three-factor model because it treated a single item (item 5) separately, aligning with prior research

**Table 5. Gender-based differences in ADS-S questionnaire item scores.**

| Item | Male (n = 196), mean (SD) | Female (n = 193), mean (SD) | Mean difference (95% CI) | t-value | P |
|---|---|---|---|---|---|
| Item 1: Upsetting | 1.92 (0.87) | 2.17 (0.91) | 0.25 (0.07–0.43) | 2.751 | 0.006 |
| Item 2: Sense of control | 2.68 (1.00) | 2.68 (0.89) | 0.00 (-0.19–0.19) | 0.015 | 0.988 |
| Item 3: Uncertainty | 1.69 (0.85) | 2.03 (0.93) | 0.33 (0.16–0.51) | 3.683 | <0.001 |
| Item 4: Deterioration likelihood | 2.53 (0.79) | 2.69 (0.78) | 0.17 (0.01–0.32) | 2.098 | 0.037 |
| Item 5: Self-control | 2.21 (0.83) | 2.38 (0.80) | 0.17 (0.01–0.33) | 2.062 | 0.040 |
| Item 6: Coping | 2.79 (0.86) | 2.86 (0.68) | 0.07 (-0.08–0.23) | 0.939 | 0.348 |
| Item 7: Goal interference | 1.77 (0.89) | 2.01 (0.96) | 0.24 (0.06–0.42) | 2.562 | 0.011 |
| **Total score** | **15.59 (3.77)** | **16.82 (3.39)** | **1.23 (0.52–1.95)** | **3.379** | **0.001** |

n, number; SD, standard deviation; 95% CI, 95% confidence interval.

[11], where issues arose due to its floor effect and low correlation with the overall items. Treating a single item as a separate factor contradicts statistical modeling principles, as single items lacks necessary connections to other related items to form a separate dimension or subcomponent within a construct [20, 30]. Previous research has consistently shown that scales with multiple items have higher predictive validity compared to scales based on a single item [30].

A two-factor model, comprising factor 1 (psychological impact of diabetes) and factor 2 (sense of self-control), better fit the data compared to a one-factor structure. This was evident from the lower values of SRMR and RMSEA and the higher values of TLI and CFI [23]. This is consistent with previous studies grounded in the Lazarus and Folkman's stress coping theory, where scales such as Cognitive Appraisal of Health Scale [29] and the Appraisal of Illness Scale similarly identified a multifactorial structure of appraisal [31].

Nevertheless, item 5 does not align with either factor 1 or factor 2. Japanese validation suggested it might indicate a third dimension related to the efforts made towards managing symptoms [10], which could relate to the secondary appraisal of diabetes [28]. Considering the questionnaire's origin in 1991 [9] and the recent advances in diabetes management [4–7], enhancing the ADS questionnaire could be beneficial. This improvement could involve adding items that would capture different aspects of symptom management efforts (i.e., secondary appraisal) such as regular BG monitoring, medication adherence, physical activity, dietary choices, symptom tracking, seeking information and education about managing diabetes, self-foot checks, support groups participation, and regular visits to healthcare providers [32, 33]. This ensures that the questionnaire remains relevant and reflects current diabetes management practices.

Next, the questionnaire has exhibited strong criterion validity, as evidenced by its significant correlations with the EQ-5D utility score, the EQ-VAS score, all EQ-5D dimensions, and poor HbA1c status. These results are consistent with previous studies that have also shown significant correlations between ADS and key parameters, such as glycemic control [10, 34, 35], the EQ-5D utility score [36, 37], as well as psychosocial adaptation [34, 38], the psychological general well-being index [36, 37], the 36-item generic quality of life measure [39], and the diabetes specific quality of life questionnaire [11].

Test-retest reliability was found to be good with an ICC close to 0.9. This high degree of agreement between the initial and subsequent measurements underscores the reliability and stability of the ADS-S results over time and has also been reported in previous studies [9, 10].

In assessment of discriminant validity, no significant correlation was found between the ADS-S score and age, while a weak significant correlation was observed with female gender. These findings indicate that the ADS-S questionnaire effectively distinguishes between diabetes appraisal and age, but some questionnaire aspects may overlap or vary with gender-related factors. The association of ADS-S score with female gender is expected as previous research has shown that women are more likely to perceive diabetes as having negative impact on their lives and express more anxiety and worry about the disease than men [40, 41].

Women in our sample scored significantly higher on the ADS-S questionnaire and had significantly higher levels of depression/anxiety compared to men. Furthermore, women reported significantly higher levels of distress related to their diabetes, expressed greater uncertainty about their condition, demonstrated an increased likelihood of deterioration, exhibited lower self-control in managing diabetes, and experienced more interference with their life goals compared to men. This may shed light on the reasons for the observed common differences. Earlier studies lacked discriminant validity assessment and initially focused on men [9], making direct comparisons difficult. However, subsequent Japanese [10] and Korean [11] versions confirmed the reliability and validity of ADS for both male and female populations with

T2D. More in-depth research is needed to fully understand the extent of this association and its potential implications for interpreting survey results within specific gender contexts.

## Implications for practice and research

The results of this study show that the ADS-S questionnaire is a valid and reliable tool for assessing the psychological impact and sense of self-control in Slovenian individuals with T2D. The questionnaire could be a valuable addition for healthcare professionals who want to understand the emotional and mental well-being of their patients with T2D. In addition, its high stability over time makes it a valuable tool for longitudinal studies. However, it is important to note that the questionnaire, with its current reliability characteristics, can only distinguish between two extreme groups of people - those with very low and those with very high diabetes appraisal. As there are no set cut-off points defining what is a high or low diabetes appraisal, this limitation is not significant for clinical practice. Furthermore, future research should explore adding additional items to the questionnaire to better capture the secondary appraisal of diabetes and explore the influence of female gender on ADS scores.

## Strengths and weaknesses

The strength of this study lies in its comprehensive approach to assessing the psychometric properties of the Slovenian version of the ADS questionnaire in the reasonable representative sample of the Slovenian population from multiple centers. The study explores the possibility of a multidimensional structure for the ADS questionnaire, which is important for understanding the complexity of diabetes appraisal and discuss future questionnaire improvements. However, the questionnaire itself raised some concerns regarding several items having floor effects and potential presence of item redundancy or incompleteness. Furthermore, the study used convenience sampling among participants during routine visits to healthcare centers, which may introduce the selection bias as those seeking care may have different attitudes and experiences than those who do not. Lastly, there is no available data on response rate.

## Conclusions

As the prevalence of T2D continues to rise, the timely identification of individuals requiring additional support in managing T2D is critical. Given the limited availability of validated Slovenian diabetes-specific quality of life questionnaires, the ADS-S questionnaire proves invaluable for assessing psychological impact and self-control in Slovenian T2D patients. Integrating this questionnaire into routine clinical practice offers several advantages, including rapid identification of individuals who may be at risk of not adhering to their treatment plan and experiencing psychological distress. Early intervention and personalized care can empower patients to overcome psychological barriers and improve health outcomes. On the other hand, the multidimensional structure of the ADS-S questionnaire suggests the need for a more nuanced approach to understanding appraisal of diabetes and points to potential avenues for further research and refinement to enable a more comprehensive assessment of patient experience in the future.

## Supporting information

**S1 Appendix. English version of the Appraisal of Diabetes Scale (ADS).**
(DOCX)

**S2 Appendix. Slovenian version of the Appraisal of Diabetes Scale (ADS-S).**
(DOCX)

## Author Contributions

**Conceptualization:** Matic Mihevc, Špela Miroševič, Majda Mori Lukančič, Tina Virtič Potočnik, Črt Zavrnik, Marija Petek Šter, Zalika Klemenc-Ketiš, Antonija Poplas Susič.

**Data curation:** Matic Mihevc, Špela Miroševič, Majda Mori Lukančič, Tina Virtič Potočnik, Črt Zavrnik.

**Formal analysis:** Matic Mihevc, Špela Miroševič.

**Funding acquisition:** Zalika Klemenc-Ketiš, Antonija Poplas Susič.

**Investigation:** Matic Mihevc, Majda Mori Lukančič.

**Methodology:** Matic Mihevc, Špela Miroševič.

**Project administration:** Matic Mihevc.

**Resources:** Matic Mihevc, Majda Mori Lukančič, Zalika Klemenc-Ketiš, Antonija Poplas Susič.

**Software:** Matic Mihevc, Špela Miroševič.

**Supervision:** Matic Mihevc, Marija Petek Šter, Zalika Klemenc-Ketiš, Antonija Poplas Susič.

**Validation:** Matic Mihevc, Špela Miroševič.

**Visualization:** Matic Mihevc.

**Writing – original draft:** Matic Mihevc.

**Writing – review & editing:** Matic Mihevc, Špela Miroševič, Majda Mori Lukančič, Tina Virtič Potočnik, Črt Zavrnik, Marija Petek Šter, Zalika Klemenc-Ketiš, Antonija Poplas Susič.

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
