## [Decision Letter · Decision Letter 0]

16 Jan 2024

PONE-D-23-36026Assessing the Reliability and Validity of the Slovenian Version of the Appraisal of Diabetes Scale (ADS-S) in Type 2 Diabetes PatientsPLOS ONE

Dear Dr. Mihevc,

Thank you for submitting your manuscript to PLOS ONE. After careful consideration, we feel that it has merit but does not fully meet PLOS ONE’s publication criteria as it currently stands. Therefore, we invite you to submit a revised version of the manuscript that addresses the points raised during the review process.

We look forward to receiving your revised manuscript.

Kind regards,

Gregor Stiglic, Ph.D.

Academic Editor

PLOS ONE

“The research is financially supported by the SCUBY project, an international research initiative co-financed by the European Union through the H2020 - Health Program (H2020-SC1) and identified by contract number 825432 - SCUBY. SM acknowledges support from the Slovenian Research Agency ARIS (program P3-0339).”

3. In the online submission form you indicate that your data is not available for proprietary reasons and have provided a contact point for accessing this data. Please note that your current contact point is a co-author on this manuscript. According to our Data Policy, the contact point must not be an author on the manuscript and must be an institutional contact, ideally not an individual. Please revise your data statement to a non-author institutional point of contact, such as a data access or ethics committee, and send this to us via return email. Please also include contact information for the third party organization, and please include the full citation of where the data can be found.

4. Please ensure that you include a title page within your main document. You should list all authors and all affiliations as per our author instructions and clearly indicate the corresponding author.

Reviewers' comments:

Reviewer's Responses to Questions

**Comments to the Author**

1. Is the manuscript technically sound, and do the data support the conclusions?

Reviewer #1: Yes

Reviewer #2: Yes

2. Has the statistical analysis been performed appropriately and rigorously? 

Reviewer #1: Yes

Reviewer #2: Yes

3. Have the authors made all data underlying the findings in their manuscript fully available?

Reviewer #1: Yes

Reviewer #2: Yes

4. Is the manuscript presented in an intelligible fashion and written in standard English?

Reviewer #1: Yes

Reviewer #2: Yes

5. Review Comments to the Author

Reviewer #1: The article is well written. The summary and statistics related to the validity and reliability of the tool are well done with the appropriate statistical analyses. The only section I had a question about was the 3.7 discrimination validity, however my questions were addressed in the discussion section. There may be cultural differences.

Reviewer #2: The work is written in an appropriate and orderly manner.

I do not have any suggestions other than the suggestion to be added only in the summary section.

The number of samples and why so many people were included can be added to the method section of the summary.

6. PLOS authors have the option to publish the peer review history of their article (what does this mean?). If published, this will include your full peer review and any attached files.

Reviewer #1: No

Reviewer #2: No

---

## [Author Response · Author response to Decision Letter 0]

22 Feb 2024

Dear Editor-in-chief, 

Please find below the response to the comments of the reviewers and technical editors. Firstly, we would like to thank the editor and reviewers for the valuable comments which helped us improve our article. We revised and included corrections according to the reviewers’ comments to our best knowledge. Additionally, we made some grammar changes to improve readability. We hope that our article will now be suitable for publication in the PLOS ONE journal. 

RESPONSE TO JOURNAL REQUIREMENTS 

Answer: Thank you for your comment. We have adjusted the manuscript's style in accordance with the provided styling guidelines.

2. Thank you for stating the following financial disclosure: “The research is financially supported by the SCUBY project, an international research initiative co-financed by the European Union through the H2020 - Health Program (H2020-SC1) and identified by contract number 825432 - SCUBY. SM acknowledges support from the Slovenian Research Agency ARIS (program P3-0339).” Please state what role the funders took in the study. If the funders had no role, please state: "The funders had no role in study design, data collection and analysis, decision to publish, or preparation of the manuscript."

Answer: Thank you for your comment. We have incorporated the suggested statement into our cover letter. Additionally, we kindly request your assistance in updating the online submission form accordingly.

3. In the online submission form you indicate that your data is not available for proprietary reasons and have provided a contact point for accessing this data. Please note that your current contact point is a co-author on this manuscript. According to our Data Policy, the contact point must not be an author on the manuscript and must be an institutional contact, ideally not an individual. Please revise your data statement to a non-author institutional point of contact, such as a data access or ethics committee, and send this to us via return email. Please also include contact information for the third-party organization, and please include the full citation of where the data can be found.

Answer: Thank you for your comment. We have successfully uploaded the dataset to the Dataverse platform, making it freely accessible online at: https://dataverse.nl/dataset.xhtml?persistentId=doi:10.34894/SPGBBR. 

4. Please ensure that you include a title page within your main document. You should list all authors and all affiliations as per our author instructions and clearly indicate the corresponding author.

Answer: Thank you for your comment. We have incorporated the title page into the main document, and the styling has been adjusted in accordance with the guidelines specified by the journal.

Answer: Thank you for your comment. We included ethic statement only in the methods section. 

6. Please review your reference list to ensure that it is complete and correct. /…/ 

Answer: Thank you for your comment. We reviewed reference list again. 

RESPONSE TO REVIEWERS 

1. Is the manuscript technically sound, and do the data support the conclusions?

Reviewer #1: Yes

Reviewer #2: Yes

2. Has the statistical analysis been performed appropriately and rigorously?

Reviewer #1: Yes

Reviewer #2: Yes

3. Have the authors made all data underlying the findings in their manuscript fully available?

Reviewer #1: Yes

Reviewer #2: Yes

4. Is the manuscript presented in an intelligible fashion and written in standard English?

Reviewer #1: Yes

Reviewer #2: Yes

5. Review Comments to the Author

Reviewer #1: The article is well written. The summary and statistics related to the validity and reliability of the tool are well done with the appropriate statistical analyses. The only section I had a question about was the 3.7 discrimination validity, however my questions were addressed in the discussion section. There may be cultural differences.

Answer: Thank you for your comment. We appreciate the positive feedback. 

Reviewer #2: The work is written in an appropriate and orderly manner. I do not have any suggestions other than the suggestion to be added only in the summary section. The number of samples and why so many people were included can be added to the method section of the summary.

Answer: Thank you for your comment. We appreciate the positive feedback. We have restructured the summary in alignment with the latest journal requirements, which advocate a non-structured abstract. Additionally, as per your suggestion, we have incorporated the number of samples into the summary section.

---

## [Editor Report · Decision Letter 1]

6 Mar 2024

Assessing the Reliability and Validity of the Slovenian Version of the Appraisal of Diabetes Scale (ADS-S) in Type 2 Diabetes Patients

PONE-D-23-36026R1

Dear Dr. Mihevc,

We’re pleased to inform you that your manuscript has been judged scientifically suitable for publication and will be formally accepted for publication once it meets all outstanding technical requirements.

Kind regards,

Gregor Stiglic, Ph.D.

Academic Editor

PLOS ONE
---

## [Editor Report · Acceptance letter]

15 Mar 2024

PONE-D-23-36026R1 

PLOS ONE

Dear Dr. Mihevc, 

I'm pleased to inform you that your manuscript has been deemed suitable for publication in PLOS ONE. Congratulations! Your manuscript is now being handed over to our production team.

Kind regards, 

on behalf of

Dr. Gregor Stiglic 

Academic Editor

PLOS ONE